# Integrating Molecular Diagnostics and GIS Mapping: A Multidisciplinary Approach to Understanding Tuberculosis Disease Dynamics in South Africa Using Xpert MTB/RIF

**DOI:** 10.3390/diagnostics13203163

**Published:** 2023-10-10

**Authors:** Lesley Erica Scott, Anne Nicole Shapiro, Manuel Pedro Da Silva, Jonathan Tsoka, Karen Rita Jacobson, Michael Emch, Harry Moultrie, Helen Elizabeth Jenkins, David Moore, Annelies Van Rie, Wendy Susan Stevens

**Affiliations:** 1Wits Diagnostic Innovation Hub, Faculty of Health Science, University of the Witwatersrand, Johannesburg 2093, South Africa; pedro.dasilva@nhls.ac.za (M.P.D.S.); jtsoka@ilead.org.za (J.T.); wendy.stevens@wits.ac.za (W.S.S.); 2Department of Biostatistics, Boston University School of Public Health, Boston, MA 02118, USA; anshap@bu.edu (A.N.S.); jenkins.helen@gmail.com (H.E.J.); 3National Priority Program of the National Health Laboratory Services (NHLS), Johannesburg 2131, South Africa; 4Division of Infectious Diseases, Boston Medical Center, Boston, MA 02118, USA; karen.jacobson@bmc.org; 5Department of Epidemiology, University of North Carolina School, Chapel Hill, NC 27127, USA; emch@unc.edu; 6Department of Geography and Environment, University of North Carolina, Chapel Hill, NC 27599, USA; 7National Institute for Communicable Diseases of the National Health Laboratory Service, Johannesburg 2192, South Africa; harrym@nicd.ac.za; 8Department of Clinical Research, Faculty of Infectious and Tropical Diseases, London School of Hygiene and Tropical Medicine, London WC1E 7HT, UK; david.moore@lshtm.ac.uk; 9Faculty of Medicine and Health Sciences, University of Antwerp, 2000 Antwerpen, Belgium; annelies.vanrie@uantwerpen.be

**Keywords:** Xpert MTB/RIF, mycobacterial load, rifampicin resistance, spatial analysis, tuberculosis, South Africa

## Abstract

An investigation was carried out to examine the use of national Xpert MTB/RIF data (2013–2017) and GIS technology for MTB/RIF surveillance in South Africa. The aim was to exhibit the potential of using molecular diagnostics for TB surveillance across the country. The variables analysed include *Mycobacterium tuberculosis* (*Mtb*) positivity, the mycobacterial proportion of rifampicin-resistant *Mtb* (RIF), and probe frequency. The summary statistics of these variables were generated and aggregated at the facility and municipal level. The spatial distribution patterns of the indicators across municipalities were determined using the Moran’s *I* and Getis Ord (Gi) statistics. A case-control study was conducted to investigate factors associated with a high mycobacterial load. Logistic regression was used to analyse this study’s results. There was striking spatial heterogeneity in the distribution of *Mtb* and RIF across South Africa. The median patient age, urban setting classification, and number of health care workers were found to be associated with the mycobacterial load. This study illustrates the potential of using data generated from molecular diagnostics in combination with GIS technology for *Mtb* surveillance in South Africa. Spatially targeted interventions can be implemented in areas where high-burden *Mtb* persists.

## 1. Introduction

Tuberculosis (TB) remains a global health concern, with reports of 1.6 million deaths in 2021 [1]. In 2020, the WHO updated the list of 30 high TB burden countries (HBCs), which account for 86% of all estimated incident TB cases worldwide [2]. Among the HBCs, South Africa has the third highest absolute number of reported cases (172,200) and the fifth highest number of estimated prevalent TB cases (304,000) [1]. It also has the largest number of HIV-associated TB cases (81,800) and the second-largest number of diagnosed multidrug-resistant (MDR)-TB cases (7100) [1].

Despite South Africa’s high TB burden, relatively little is known about the spatial heterogeneity of tuberculosis within the country. South Africa has a centrally managed laboratory database well suited for surveillance purposes, especially after the implementation of the Xpert MTB/RIF (Xpert; Cepheid, Sunnyvale, CA, USA) molecular test in 2011 [3]. The COVID-19 pandemic catapulted the use of molecular diagnostics to indicate changes in the disease profile; South Africa leveraged its laboratory database to introduce the concept of using qualitative diagnostic values at a national level for real-time pandemic surveillance [4]. Previous studies have also demonstrated the potential to monitor and assess TB and drug-resistant TB disease dynamics using national surveillance data [5,6,7,8]. A 2018 retrospective spatial analysis of routinely collected laboratory data in South Africa’s Western Cape province revealed significant spatial and temporal heterogeneity in rifampicin-resistant (RR) TB [9]. However, as these data pre-dated the implementation of Xpert, they lack crucial qualitative diagnostic variables that aid in monitoring TB and RR TB. These studies highlight the urgent need for more granular surveillance to identify the hotspots of transmission and link them to the possibility of targeting case-finding strategies in areas of concentrated risk [8].

A geographical analysis of South Africa’s routinely collected TB laboratory tests via Geographic Information System (GIS) methods at the national level enables the spatiotemporal monitoring of the epidemic across the entire population, providing timely data and spatially targeted interventions for disease control [10]. GIS methods are a powerful tool that allow for the analysis of multiple layers of data, seamlessly integrating diverse datasets such as population demographics, healthcare infrastructure, and environmental factors. This comprehensive approach enhances our understanding of the complex dynamics of the TB epidemic and supports evidence-based decision making. This study is an initial analysis of the South African National Priority Program (NPP) of the National Health Laboratory Service’s (NHLS) TB data with the aim to (1) analyse the collected data from 2013 through 2016 using GIS methods for identifying high risk areas that require interventions and (2) show the prospects for national laboratory data use. We analyse centrally collected national Xpert laboratory data over a four-year period (2013–2016) with GIS and other mapping tools to exemplify the value of the aggregated Xpert molecular data in TB control and perform a municipal-level investigation of national data and a facility-level analysis of data from South Africa’s Eastern Cape province to highlight the applications of the geographical analysis of molecular laboratory diagnostics for TB surveillance at a small-area scale. Using spatial results from the analysis of the Eastern Cape province, we also performed a case-control study to identify factors associated with a high TB burden load. Despite ending in 2016, these data provide the first opportunity to demonstrate the benefits of multidisciplinary analyses involving geospatial methods for TB surveillance.

## 2. Materials and Methods

### 2.1. Study Setting

Prior to a 2016 redistricting, South Africa was divided into 9 provinces, 52 districts, and 234 municipalities. Xpert was rolled out in 2011. By 2013, over 5000 healthcare facilities across all municipalities and provinces were connected to 300 GeneXpert (Cepheid, Sunnyvale, CA, USA) instruments installed in 193 national laboratories [11]. The distribution of GeneXpert instruments and health care facilities is representative of the population density.

At the time of the national implementation of Xpert’s roll out, the NPP ensured each GeneXpert instrument was interfaced with the laboratory information system (LIS) so that all authorised test results were available electronically (and as a hard copy) to treating health practitioners [10]. The Xpert assay is a novel, fully automated polymerase chain reaction (PCR)-based procedure that can rapidly diagnose the presence of *Mycobacterium tuberculosis* (*Mtb*) directly in sputum with high sensitivity and specificity [12]. The GX4.7b assay software generates the cycle-threshold (Ct) value [12,13,14], which is a measure of the *Mtb* burden [15] in tested specimens and could replace smear microscopy status as a marker of infectiousness, especially in high HIV burden settings [16]. The Xpert assay determines RR based on mutations in the 81-base-pair (bp codons 507–533) regions of the β-subunit of *rpoB*, the RNA polymerase enzyme, using five overlapping probes. RR is determined based on any mutation in at least one of the five probes. The probes involved in detection are characterized as probe A (codons 507–511), probe B (codons 511–518), probe C (codons 518–523), probe D (codons 523–529), and probe E (codons 529–533) [17]. Each probe corresponds to a different mutation. Mutations in these regions contribute to 93% of the RR [18].

### 2.2. Data Sources

All Xpert test data results were queried from the NHLS’s Corporate Data Warehouse (CDW) through two LISs: DisaLab (Laboratory System Technologies, Johannesburg, South Africa) and TrakCare (InterSystems, Cambridge, MA, USA). A total of 11,345,104 Xpert test records were available for the study period (2013–2017). This study period was selected to minimize variability during the early implementation phase (2011–2013) and latter (post September 2017) phase, during which the Xpert was transitioned to Xpert MTB/RIF Ultra (Cepheid).

We conducted a case-control study in the Eastern Cape province to assess factors associated with a high facility mycobacterial load. We developed a methodological framework using GIS and the Xpert test data to identify and select high- (case) and low- (control) burden facilities to be sampled out of the total of 958 facilities in the study area. First, given that TB transmission does not stop at the borders of census areas or districts, we used a spatial and spatio-temporal analysis to identify the most appropriate geographical units of analysis. Secondly, because the public health infrastructure plays a crucial role in targeted interventions to break the TB transmission cycle, we used a hotspot analysis of the mycobacterial load on the primary care clinics. The sampled cases and controls were 32 and 27 clinics with the highest and the lowest mycobacterial load in the study area, respectively. High-burden facilities had median mycobacterial loads of 4.48 log 10 CFU/mL or greater, and low-burden facilities had median mycobacterial loads under 4.48 log 10 CFU/mL. Data were collected through observation, interviews with clinic management and staff, and a review of clinic registries and statistics. At the patient level, we used a convenience sampling strategy to identify 12 patients to be interviewed at each selected clinic. Identifying factors measured in the survey was based on the theory proposed by Link and Phelan [19] and Clouston and Link [20]. This approach has been widely employed in non-TB studies and was adapted for this study, as shown in Appendix A. The information collected included issues related to socio-economic conditions, health-seeking behaviour, lack of skilled workers, quality of care at public facilities, accessibility of care facilities (distance and opening hours), attitudes towards TB and HIV, and treatment adherence. Participants were assigned a study number as to avoid the use of personally identifying data.

The GIS shapefile data was downloaded from the 2016 repository of the Municipality Demarcation Board (MDB) of South Africa [21]. The population estimates were extracted from the most recent 2016 census online database of Statistics South Africa [22].

### 2.3. Data Preparation

The Ct value is the number of PCR cycles required to amplify mycobacterial DNA above a background threshold [23]. Ct values can also be used to quantify the amount of *Mycobacterium tuberculosis* (*Mtb*) in a sputum specimen (henceforth, “mycobacterial load”), a measure of the force of infection, using the linear relationship between Ct and colony-forming units (CFU/mL) [14,23,24]. Based on the internal calibration of GeneXpert’s instruments, the relationship between the Ct value and CFU/mL were defined by Equation (1):*Ct* = −3log (CFU/mL) + 35,(1)

For simplicity, the CFU/mL was expressed as a log scale, as shown in Equation (2).
(2)$$\log\;\left(\mathrm{CFU}/\mathrm{mL}\right)=\;\frac{Ct-35}{-3}\times 100,$$

The median mycobacterial load, total number of tests, number of tests positive for *Mtb*, and number of tests positive for RR were calculated at the facility level and aggregated up to the municipal and provincial level. *Mtb* positivity was calculated by dividing the number of positive tests by the total number of tests performed. RR positivity was calculated by dividing the number of tests positive for RR by the number of tests positive for *Mtb.*

The frequency of each probe was calculated as the total number of indicated mutations in each probe (A, B, C, D, or E) divided by the total number of tests positive for RR. Aggregated data were then merged with local municipality GIS shapefiles for analysis.

### 2.4. Analysis

#### 2.4.1. Mapping and Spatial Analysis

Spatial autocorrelation, the degree of similarity of an event over a geographical surface [25], was investigated using the Moran’s *I* statistic to measure the presence, strength, and direction of spatial dependency and association among municipalities [26]. The local Moran’s *I* is denoted by Equation (3):(3)$$I_{i}=\frac{x_{i}-\underline{x}}{m_{2}}\sum_{j=1}^{N}w_{ij}\;\left(x_{j}-\underline{x}\right),$$

The Moran’s *I* statistic was applied to the variables *Mtb* positivity, RR positivity, mycobacterial load, and probe frequency. The *p*-value and z-score were used to evaluate the significance of the Moran’s *I* test statistic. A disease is considered spatially aggregated and statistically clustered when the Moran’s *I* is > 0 and with a z-score of ≥ 1.96 [27].

Hotspots (areas with an elevated rate of a particular variable [28]) and cold spots (areas with a lower rate of a particular variable) were investigated using the Getis Ord (Gi) statistic, which identifies the location, significance, and type (cold and hotspots) of cluster [27]. The Gi statistic estimates neighbouring municipalities geographically aggregated with similar rates. The Gi statistic was calculated for each local municipality for the *Mtb* positivity, RR positivity, mycobacterial load, and probe frequency.

#### 2.4.2. Logistic Regression

Logistic regression was conducted to determine the association between the predictor variables and the outcome variable (low or high mycobacterial load). The predictors are related to the outcome variables by Equation (4):*Logit*(*p*) = *β*_0_ + *β*_1×1_ + ··· + *β_p_X_p_*,(4)
where *p* denotes the probability of high mycobacterial load, *β*_1_ is the coefficient of the predictor *X*_1_. Exp(*β*_1_) denotes the odd ratio, which is used to interpret the effect of the predictor variable on the outcome variable. We used the first step of the purposeful variable selection process as discussed by Bursac et al. [29] to select significant variables at a 10% significance level in univariate models, and these variables were considered potential candidates for a multivariate analysis. We then ran the backward elimination method of variable selection. The final model is reported with the parameter estimates of the selected variables.

### 2.5. Software

Data pre-processing and risk analysis were performed in *R* version 4.1.3. Spatial data analyses were performed using ArcGIS version 10.6.

### 2.6. Ethics

Approval to access the laboratory test results data was obtained from the University of the Witwatersrand, Johannesburg, South Africa’s Human Research Ethics Committee (M160978) and the NHLS through the NPP data access policy termed ILDAC (Integrated Laboratory Data Analytics for Care). Permission to conduct the case-control study was obtained from the University of the Witwatersrand Human Research Ethics Committee clearance committee (approval no M15021).

## 3. Results

### 3.1. Spatial and Temporal Distribution of Mtb Testing

Nationally, the volume of *Mtb* testing follows a yearly cycle that peaks around September and reaches a minimum in December of each year. *Mtb* positivity is inversely proportional to the volume of *Mtb* tests, although the peaks of test positivity lower each year (Figure 1). A total of 11,345,104 Xpert tests were performed during the study period of 2013–2017. Of those, 1,124,995 (9.9%) were positive for *Mtb.* The Western Cape province had the highest *Mtb* positivity (14.8%), followed by the Northern Cape province (11.1%). In contrast, the Limpopo (5.7%) and the KwaZulu Natal (8.9%) provinces had the lowest rates of *Mtb* positivity (Table 1).

The Moran’s *I* statistic (I = 0.46, *p* < 0.001) indicated a significant clustering of *Mtb* positivity in municipalities. The analysis revealed 36 municipalities as hotspots across the Northern Cape, Eastern Cape, and Western Cape provinces. Conversely, 41 municipalities were identified as cold spots in the provinces of Limpopo, KwaZulu Natal, Eastern Cape, and North West based on the Gi statistic (Figure 2). The analysis of the force of infection, measured by the mycobacterial load, revealed significant spatial clustering among municipalities (I = 0.53, *p* < 0.001). The hotspot analysis of the mycobacterial load revealed 42 municipalities in the Western Cape, Northern Cape, and Eastern Cape provinces with elevated levels of mycobacterial load, and 48 municipalities in the Gauteng, KwaZulu Natal, Eastern Cape, and Free State provinces exhibited cold spots, indicating areas of a lower mycobacterial load during the study period (Figure 2). See Appendix A for the province locations.

### 3.2. Spatial and Spatio-Temporal Analyses of Eastern Cape

The median mycobacterial load pooled across the study period showed significant spatial heterogeneity at the facility level in the Eastern Cape (I = 0.42, *p* < 0.001) (Figure 3). A higher median mycobacterial load is correlated with a higher volume of *Mtb* testing per 100,000 population.

Figure 4 shows facility mycobacterial load patterns from 2013–2016. Hotspot clusters were primarily in the southwest, and coldspot clusters were in the northeast. Facilities within the hotspot cluster displayed consistently high mycobacterial loads compared to neighbouring facilities, while facilities within the coldspot cluster displayed consistently lower mycobacterial loads compared to neighbouring facilities. The prominent hotspot shifted eastward during the study period. It shrunk in size between 2013 and 2014 and then expanded again in 2016. The coldspot cluster shifted south-eastward during the study period, and its size decreased over time.

### 3.3. Factors Associated with High Mycobacterial Load

This study revealed that only 25% of the facilities had health care workers (HCWs) that are doctors; however, almost all the facilities had nurses and non-specified HCWs (HCWs that are not doctors or nurses). Most (93%) of the facilities surveyed are primary healthcare facilities, while 7% are community health centres. The findings further show that about 46% of the facilities had not been optimised and evaluated for TB-infection control over the past five years. At the patient level, it was observed that the average age of the patients tested is 42 ± 10 years. The distances of patients’ journeys to facilities ranged from 1 to 7 km, with a mean of 2 km, and most of them travel by taxi and train, with an average cost of ZAR 20 (range: 0 to 100).

At the 10% level of significance, evidence of association with a high mycobacterial load is suggested for the following variables: median patient age, number of nurses, number of non-specified HCWs, population count, and land use class (urban/rural). See Appendix A for results from all univariate models. After running backwards selection, we are left with three variables that are all significantly related to a high mycobacterial load: median patient age, number of non-specified HCWs, and land use class. The results from the final model are shown in Table 2. A one-year increase in the median patient age is associated with a 14% decrease in the odds of a higher mycobacterial load, although this result is borderline insignificant (95% Confidence Interval (CI) 0.82, 1.02)). An increase in the number of non-specified HCWs in a facility is associated with a decrease in the odds of having a higher mycobacterial load (odds ratio (OR) = 0.69, 95% CI = 0.46, 0.89). Facilities in urban areas have 82.4 times the odds of having high mycobacterial load (95% CI 5.09, 1334.2) compared to rural areas. We note the wide confidence interval here due to data sparsity.

### 3.4. Spatial Analysis of RR Mtb

A total of 70,596 of the Xpert tests performed during the study period reported RR, resulting in an RIF positivity of 6%. Mpumalanga had the highest rates of RIF positivity (8.4%) followed by KwaZulu Natal. The North West and Western Cape provinces showed the lowest rates of RIF positivity (5%) (Table 3). The spatial autocorrelation analysis revealed a significant clustering of municipalities with RR (I = 0.59, *p* < 0.001). Hotspots of RR were seen in 35 municipalities in the Mpumalanga and KwaZulu Natal provinces (Figure 5). Cold spots of RR were observed in municipalities in the Western Cape, North West, Free State, Northern Cape, and Eastern Cape provinces.

### 3.5. Spatial Analysis of the Five Xpert MTB/RIF Molecular Probes

Across the four-year study period, 8.1% (5730/70,596) of the RR test results had missing values due to changes in the LIS programs and required some manual imputation. The distribution of probe reporting in the Xpert laboratory test results displayed notable variations (Figure 6). Probe E emerged as the most reported with a prevalence of 56.94% (35,403/62,177) across the tested population (Figure 6). Probes D and B followed with frequencies of 19.15% (11,909/62,177) and 16.34% (10,159/62,177), respectively. In contrast, probe A and probe C were reported with much lower frequencies, representing 6.20% (3853/62,177) and 1.37% (853/62,177) of the tested specimens, respectively. Furthermore, a subset of 4.49% (2789/62,177) of the specimens reported the presence of multiple probes, indicating a potential co-occurrence or co-infection.

The spatial autocorrelation analysis of probe frequencies conferring RR revealed RR *Mtb* clusters for all probes across the municipalities (Figure 7). Significant hotspot and cold spot clusters were observed for all the five probes, all in different regions across South Africa. The Northern Cape province showed predominant hotspots of RR detected by probe A, with clusters extending to the Free State, Western Cape, and Limpopo provinces (Figure 7). In the Western Cape and Mpumalanga provinces, RR hotspots of probe E were observed. The Limpopo province showed a hotspot detected by probe D, which extended into the North West and Gauteng provinces. The Eastern Cape province showed an RR hotspot detected by probe B.

## 4. Discussion

This study aimed to exhibit how molecular diagnostic values can be used to identify areas of a high burden and risk of *Mtb* and RR in South Africa to help prioritise resources and target studies to understand the local drivers of their emergence and spread. The results revealed significant variations in the distribution of *Mtb* and RR across South Africa, with some municipalities having diagnostic rates that were more than five times higher than others. High levels of *Mtb* positivity, mycobacterial load, the proportion of RR, and their hotspots were mostly found in urban municipalities with high population densities, indicating the potential presence of micro-epidemics. This finding was echoed by the logistic regression analysis of risk factors associated with a high mycobacterial load in the Eastern Cape province, which reported significantly increased odds of a high mycobacterial load in urban areas.

Much spatial variation in the distribution of a mycobacterial load (force of infection) across different regions in South Africa was established, with the greatest force of infection observed in the Western Cape province and certain municipalities of the Northern Cape province. The greater force of infection in these regions could be attributed to the presence of recirculating *Mtb* strains and a high number of susceptible hosts due to recurrent *Mtb* infections in these municipalities [30,31]. This may be one of the drivers of geographical heterogeneity of *Mtb* burden across the country. The patterns of *Mtb* positivity and the force of infection across the country were found to be similar. Previous studies have highlighted a correlation between *Mtb* incidence and mycobacterial load in a population [32]. Therefore, municipalities with observed hotspot clusters of a mycobacterial load in the Western Cape and Northern Cape provinces, which also coincide with municipalities with high *Mtb*-positivity rates, may have also had a high *Mtb* transmission.

Socio-economic conditions associated with disease transmission, such as homelessness, the HIV epidemic, and increased migration, may have also contributed to these spatial patterns [33]. We further see this reflected in the results from our case-control study. Working-aged adults are known to have higher rates of TB [34,35]. This may be due in part to working-aged adults having difficulties accessing health services due to their work hours [36]. Furthermore, the number of unspecified healthcare workers, which can be seen as a proxy for the quality of care received, is also negatively associated with a high mycobacterial load. Further interdisciplinary analyses, such as this, are important to better understand how Ct values can be used as an indicator for monitoring TB-control practices, the uptake of treatment, and equitable healthcare access.

The frequencies of mutations varied vastly across the country with the most common *rpoB* mutations identified in the gene region 529–533, represented by probe E, followed by probes D and B, while mutations identified by probes A and C were less common. The national frequencies of mutations observed were similar to those reported in other developing countries across Africa and Asia [37,38]. Past studies also found that mutations in the region of the *rpoB* gene represented by probes E and D were the most common [39,40]. The Western Cape province had the highest detection rate of RR *Mtb* with probe E, followed by parts of KwaZulu Natal and Mpumalanga, areas where previous studies have reported high levels of drug-resistant *Mtb* [41,42,43,44]. The KwaZulu Natal and Mpumalanga provinces border Swaziland, which previously reported the unreliability of Xpert in identifying the *rpoB* 1491F mutation [45]. This may have contributed to the high rate of diagnostic RR *Mtb*, requiring targeted approaches for diagnosis and treatment.

The Eastern Cape province reported the highest detection frequency of RR *Mtb* with probe B and has almost 50% more representation than any other province, indicating the possibility of the transmission of pre-existing strains [46]. Caution, however, is needed in the interpretation of spatial associations of the probes for molecular epidemiology and inferring the transmission of drug resistance. These data report only on the single *rpoB* region, with no additional laboratory genotyping investigations performed. However, the large dataset may still be relevant for a public health approach, and future studies using additional molecular diagnostic approaches can help refine such findings.

This study highlighted the value of using connected diagnostics and GIS mapping as a surveillance tool for TB and RR in South Africa. At the national level, TB in South Africa is dynamic and geographically diverse, with the ongoing transmission of both drug-resistant and susceptible strains. Our findings identify areas in need of targeted interventions. A future data analysis could focus on specific regions at the facility level, using all available variables, including probes, and incorporating HIV laboratory and ART monitoring data, as well as results from the country’s TB drug-resistance survey, and using alternative GIS tools. This approach could be used to measure ongoing transmissions and identify gaps and effective interventions.

Centralised national data can inform national operations of laboratory coverage, monitor intervention successes, and identify near real-time gaps in services or hotspots requiring immediate action. A limitation in our study, however, is that only a single test result (Xpert MTB/RIF) is evaluated, as the linkage to smear microscopy, liquid culture, and DST results require a sophisticated algorithm-driven data linkage whose challenge is a lack of a unique patient identifier. Work to develop such an algorithm is well underway, and this study can be repeated in the future with these linked data [47,48]. Additionally, Xpert only interrogates a single gene region (*rpoB*) to identify *Mtb* and resistance to only the first-line therapy, rifampicin. Whole genome sequencing has proved useful in understanding TB transmission dynamics [49,50]. However, whole genome sequencing is costly and time intensive and not feasible to perform at a national level. The GIS analysis of Xpert is much less resource-intensive and can be used to identify smaller-scale areas that would benefit from additional sequencing. Additionally, Ct values, and thus the mycobacterial load, are known to vary at the individual level by the quality and volume of the specimen collected [51]. For this reason, they were primarily used as a qualitative variable. However, recent studies have shown the utility of Ct values for predicting smear and culture conversion [13,52]. Furthermore, due to the large amount of data used in this study, variation caused by differences in the sputum samples is negligible [53].

A final limitation is the small sample size of the case-control study. Thus, we interpret the estimates and significances with caution. Furthermore, it only shows factors that are associated with a high mycobacterial load—we cannot infer causation. That said, we consider this analysis illustrative of important interdisciplinary analyses necessary to understand factors that influence South Africa’s TB burden.

## 5. Conclusions

This study demonstrated the importance of GIS methods in analysing data from molecular diagnostics. It highlighted how centrally collected data can be used for TB surveillance, operations, and control, as well as the need for a multidisciplinary approach. Combining geographical data analytics with epidemiologic and clinical knowledge is necessary for best utilizing such data. Further research on the molecular variables discussed in this study could improve patient care and TB research in the country.

## Figures and Tables

**Figure 1 diagnostics-13-03163-f001:**
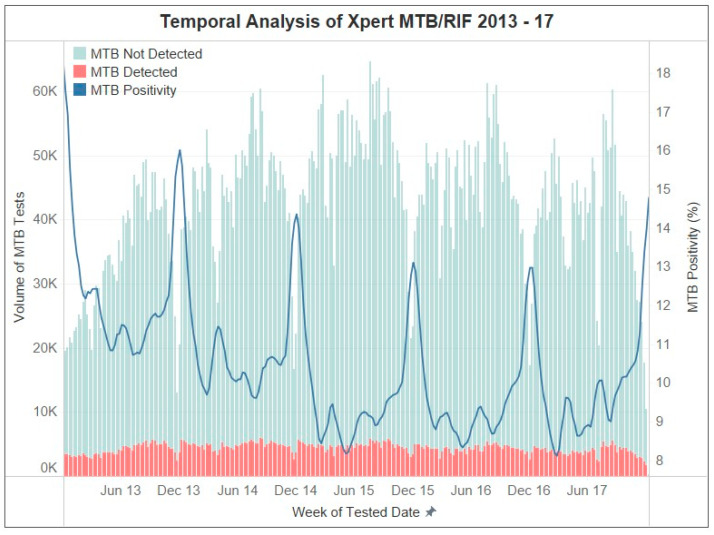
Temporal trends of Xpert MTB/RIF results between 2013 and 2017. Graphical descriptions are provided in the legends.

**Figure 2 diagnostics-13-03163-f002:**
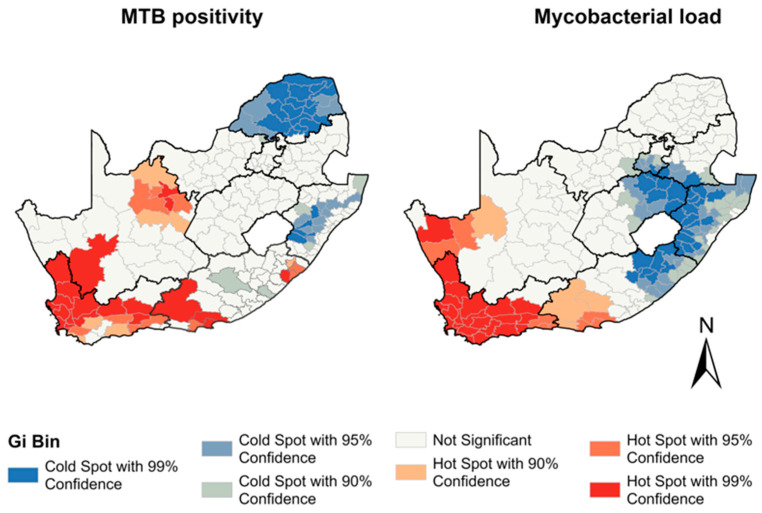
Hotspots of *Mtb* positivity and *Mtb* load between 2013 and 2017 by municipality.

**Figure 3 diagnostics-13-03163-f003:**
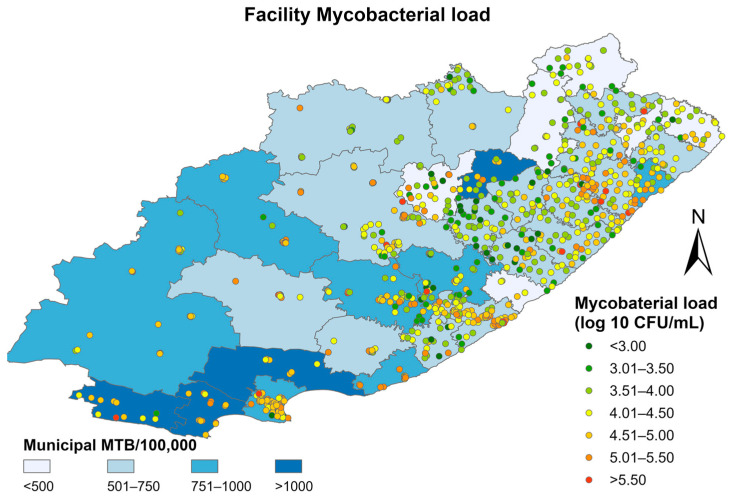
Median mycobacterial load at the facility level superimposed on municipal-level TB positivity rates between 2013–2016 in the Eastern Cape.

**Figure 4 diagnostics-13-03163-f004:**
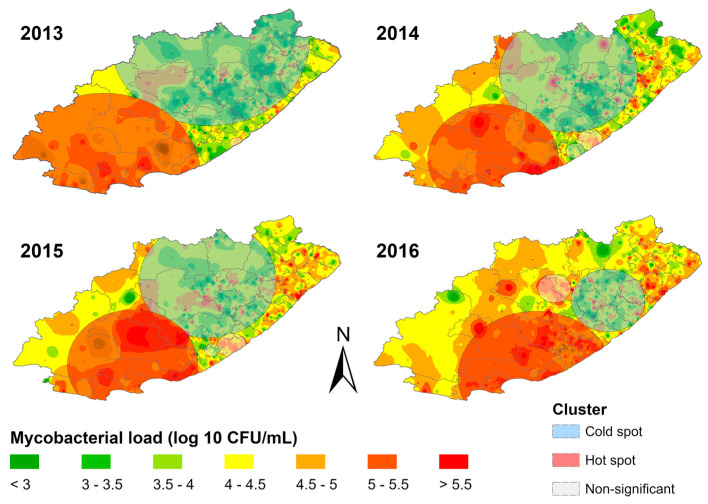
Spatial distribution of mycobacterial load clusters overlaid on inverse distance weighted interpolation layers for each study year in the Eastern Cape province.

**Figure 5 diagnostics-13-03163-f005:**
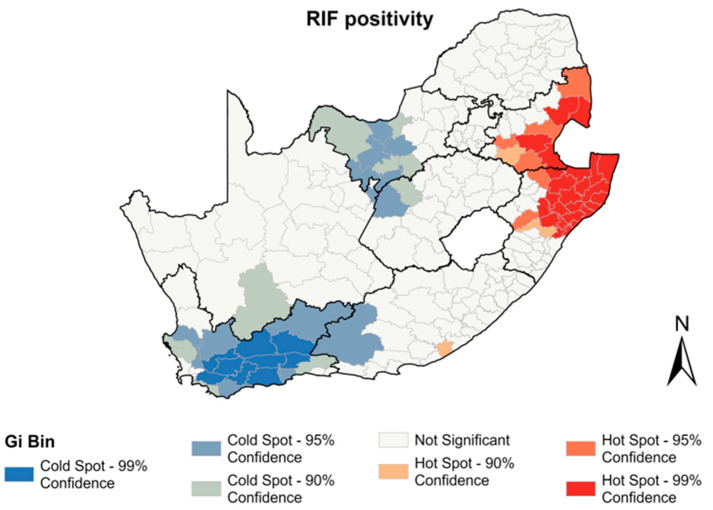
Hotspots of RR positivity between 2013–2017 by municipality.

**Figure 6 diagnostics-13-03163-f006:**
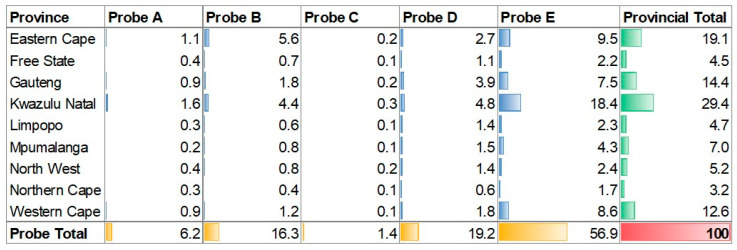
Xpert MTB/RIF probe detection frequency per province (green) and across probes (yellow). Blue is the probe frequency across the whole data set (red = 100%).

**Figure 7 diagnostics-13-03163-f007:**
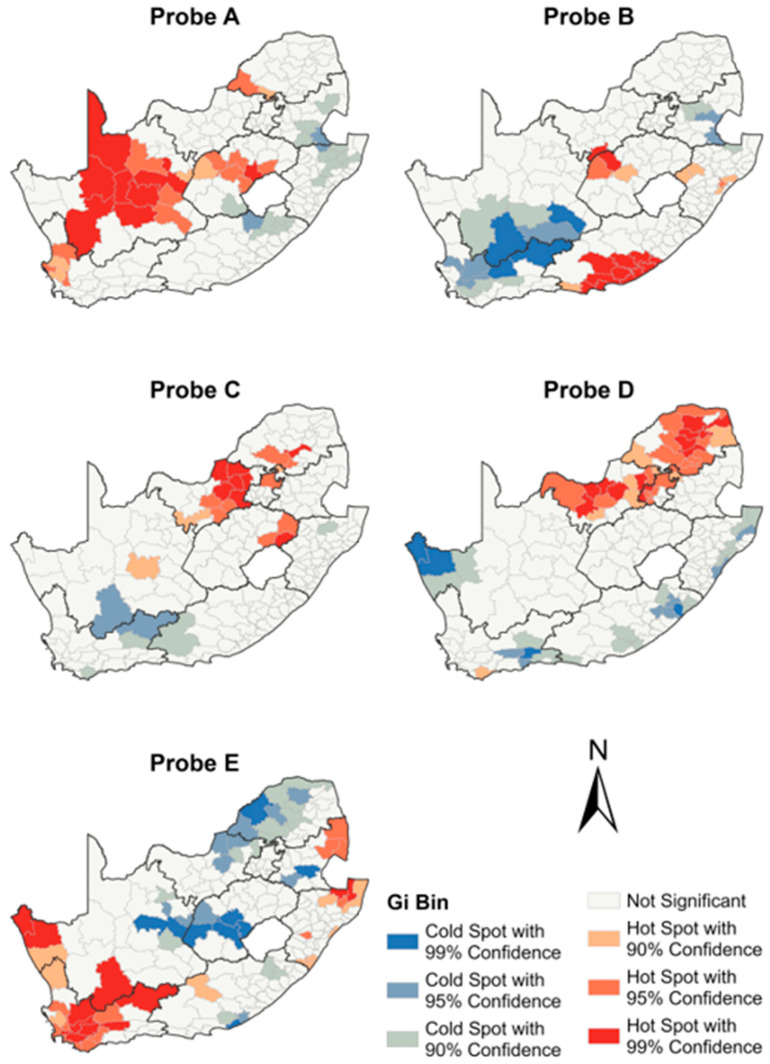
Municipal-level spatial distribution of Xpert MTB/RIF probes.

**Table 1 diagnostics-13-03163-t001:** Summary of MTB positivity rates for 2013–2017.

Province	Number of *Mtb* Tests Performed	Number of Positive *Mtb* Tests	MTB Test Positivity (%)
Eastern Cape	2,202,856	232,549	10.6
Free State	640,096	62,616	9.8
Gauteng	1,716,034	171,718	10
KwaZulu Natal	2,830,422	252,877	8.9
Limpopo	1,111,262	63,749	5.7
Mpumalanga	570,906	61,288	10.7
North West	765,694	70,987	9.3
Northern Cape	372,170	41,418	11.1
Western Cape	1,135,664	167,793	14.8

**Table 2 diagnostics-13-03163-t002:** Logistic regression analysis of risk factors associated with high mycobacterial load in the Eastern Cape (2013–2016).

Variable	95% CI	*p*-Value
Median patient age	0.86 (0.82–1.02)	0.043
Number of non-specified HCWs	0.69 (0.46–0.89)	0.007
Land use class (Urban)	82.43 (5.09–1334.23)	0.001

**Table 3 diagnostics-13-03163-t003:** Summary of RR positivity rates for 2013–2017.

Province	Number of *Mtb* Tests Performed Reporting RR	Rate of RIF Positive *Mtb* Tests (%)
Eastern Cape	13,971	6
Free State	3387	5.4
Gauteng	10,158	5.9
KwaZulu Natal	20,505	8.1
Limpopo	3363	5.3
Mpumalanga	5114	8.4
North West	3569	5
Northern Cape	2198	5.3
Western Cape	8331	5

## Data Availability

Access to primary data is subject to restrictions owing to privacy and ethics policies set by the South African Government. Requests for access to the data can be made via the Office of Academic Affairs and Research at the National Health Laboratory Service through the AARMS research project application portal: https://aarms.nhls.ac.za/, accessed on 2 September 2019.

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
