# Peer review of "Integrating Molecular Diagnostics and GIS Mapping: A Multidisciplinary Approach to Understanding Tuberculosis Disease Dynamics in South Africa Using Xpert MTB/RIF"

_diagnostics, 2023, doi:10.3390/diagnostics13203163_

Round 1

Reviewer 1 Report

I thank the authors for their efforts to do this research. However, I have the following concern.

Why this study was conducted was not clearly indicated in the introduction part.

On line 110, the authors indicated that, the study design was case control study, how and why it was case control study? To classify the clinics, the authors indicated as the “highest and the lowest mycobacterial load in the study area” … in line 118, what was the cut points for each? Despite this classification, the data analysis seems to be a cross-sectional study design as depicted in results (we can observe tables and figures). I also suggest the appropriate study design for this study was cross sectional study design as it was for the distribution. In the consecutive description for the data sources by types, it was not clear how many data were obtained from each category before it was aligned with spital locations.

Starting on line 408, authors said that they obtained informed consents from study participants despite they indicated as they used electronical data on line 102-103, it is not clear.

It needs editorial revision. E.g., at many places it said that "Error! Reference source not found.)" in line 209.

It needs editorial revision. E.g., at many places it said that "Error! Reference source not found.)" in line 209.

Author Response

Response to Reviewer 1 Comments

Point 1: Why this study was conducted was not clearly indicated in the introduction part.

Response 1: We added a sentence to last paragraph of the introduction to explain why, despite the age of the data, this study is important.

Point 2: On line 110, the authors indicated that, the study design was case control study, how and why it was case control study? To classify the clinics, the authors indicated as the “highest and the lowest mycobacterial load in the study area” … in line 118, what was the cut points for each? Despite this classification, the data analysis seems to be a cross-sectional study design as depicted in results (we can observe tables and figures). I also suggest the appropriate study design for this study was cross sectional study design as it was for the distribution. In the consecutive description for the data sources by types, it was not clear how many data were obtained from each category before it was aligned with spital locations.

Response 2: Analyses involving multiple data sources are presented in this paper. The first is a cohort study, with the cohort consisting of all individuals in South Africa who received a GeneXpert test in the public health sector. The second was a case control study conducted in South Africa's Eastern Cape Province, where we selected high burden facilities as the cases and low burden facilities as controls using GIS methods. Edits in the introduction and methods were made to clarify this distinction.

In the case-control study, each facility represents one data point. Edits were made in the methods to clarify this point. Cut-offs for high and low burden facilities have also been added.

Point 3: Starting on line 408, authors said that they obtained informed consents from study participants despite they indicated as they used electronical data on line 102-103, it is not clear.

Response 3: Electronic data was used for the cohort study consisting of all Xpert tests. Informed consent was necessary only for the case-control study. Edits made for previous comments clarify this point, and edits were made to the informed consent statement to further clarify.

Reviewer 2 Report

 The authors report the results of an investigation carried out to evaluate the use of Xpert MTB/RIF and GIS technology for MTB/RIF surveillance in South Africa. The aim is to exhibit the potentiality in using molecular diagnostics for TB surveillance across the country. They also used the Ct values to quantify the amount of Mycobacterium tuberculosis (Mtb) in sputum specimens as a predictive   measure of the force of infection. The work is well written and the methodological approach is interesting, above all because it can be exported to other medium and low income realities

Revisions to do:

1 Explains to the reader what GIS is and how it works

2 Gli autori dovrebbero sottolineare come un possibile bias del loro lavoro sia rappresentato dall’uso dei  CT per stimare, in modo puntuale, la carica batterica. In fact, the authors refer to works in which CTs were evaluated in parallel with microscopy and culture. Furthermore, these studies were carried out in different geographical areas where it is possible that the infection manifests itself with a different potentially lower microbial load and therefore the possibility cannot be excluded that it was easier to establish a linearity between: microbial load and CT.

The bias is important because using the CTs without a calibration curve and without a parameter that allows us to evaluate the quality of the collected sample (e.g. microscopic evaluation according to the Bartlett or Murray Wahingthon’s score) is a stretch. Respiratory samples can be very heterogeneous due to their viscosity and, therefore, the estimated bacterial load in a sample could be different when evaluated in different portions of the same sample.

Other studies of respiratory specimens (including a systematic review) conclude, referring to CT as a proxy for microbial count: "no universal conclusion could be reached"

Author Response

Response to Reviewer 2 Comments

Point 1: Explains to the reader what GIS is and how it works

Response 1: A sentence explaining GIS methods are used for geographic analysis was added to the introduction. Further clarifications were made to the introduction to help underscore the uses of GIS.

Point 2: The authors should underline how a possible bias of their work is represented by the use of CTs to accurately estimate the bacterial load. The bias is important because using the CTs without a calibration curve and without a parameter that allows us to evaluate the quality of the collected sample (e.g. microscopic evaluation according to the Bartlett or Murray Wahingthon’s score) is a stretch. Respiratory samples can be very heterogeneous due to their viscosity and, therefore, the estimated bacterial load in a sample could be different when evaluated in different portions of the same sample.

Other studies of respiratory specimens (including a systematic review) conclude, referring to CT as a proxy for microbial count: "no universal conclusion could be reached"

Response 2: We state on line 145-146 that the equation used relating Ct to bacterial load comes from an internal calibration.

Due to the large sample size here and the fact that we are using Ct value at an aggregate population level, rather than the individual level, by law of large numbers the variance from the bias will tend to 0. We added this as a limitation to the discussion.